# A Unified framework for randomized smoothing based certified defenses

## Abstract

Randomized smoothing, which was recently proved to be a certified defensive technique, has received considerable attention due to its scalability to large datasets and neural networks. However, several important questions still remain unanswered in the existing frameworks, such as (i) whether Gaussian mechanism is an optimal choice for certifying $\ell_2$-normed robustness, and (ii) whether randomized smoothing can certify $\ell_\infty$-normed robustness (on high-dimensional datasets like ImageNet). To answer these questions, we introduce a *unified* and *self-contained* framework to study randomized smoothing-based certified defenses, where we mainly focus on the two most popular norms in adversarial machine learning, *i.e.,* $\ell_2$ and $\ell_\infty$ norm. We answer the above two questions by first demonstrating that Gaussian mechanism and Exponential mechanism are the (near) optimal options to certify the $\ell_2$ and $\ell_\infty$-normed robustness. We further show that the largest $\ell_\infty$ radius certified by randomized smoothing is upper bounded by $O(1/\sqrt{d})$, where $d$ is the dimensionality of the data. This theoretical finding suggests that certifying $\ell_\infty$-normed robustness by randomized smoothing may not be scalable to high-dimensional data. The veracity of our framework and analysis is verified by extensive evaluations on CIFAR10 and ImageNet.

## 1 Introduction

The past decade has witnessed tremendous success of deep learning in handling various learning tasks like image classification (Krizhevsky et al., 2012), natural language processing (Cho et al., 2014), and game playing (Silver et al., 2016). Nevertheless, a major unresolved issue of deep learning is its vulnerability to adversarial samples that are almost indistinguishable from natural samples to humans but can mislead deep neural networks (DNNs) to make wrong predictions with high confidence (Szegedy et al., 2013; Goodfellow et al., 2014). This phenomenon, referred to as adversarial attack, is considered to be one of the biggest threats to the deployment of many deep learning systems. Thus, a great deal of effort has been devoted to developing defensive techniques for it. However, the majority of the existing defenses are of heuristic nature (*i.e.,* without any theoretical guarantees), implying that they may be ineffective against stronger attacks. Recent works (He et al., 2017; Athalye et al., 2018; Uesato et al., 2018) have confirmed this concern, and showed that most of those heuristic defenses actually fail to defend stronger adaptive attacks. This forces us to shift our attentions to certifiable defenses as they can classify all the samples in a predefined neighborhood of the natural samples with *a theoretically-guaranteed error bound*. Among all existing certifiable defensive techniques, randomized smoothing emerges as the most popular one due to its scalability to large datasets and arbitrary networks. Remarkably, using the Gaussian mechanism for randomized smoothing, Cohen et al. (2019) successfully certify $49\%$ accuracy on the original ImageNet dataset under adversarial perturbations with $\ell_2$ norm less than $0.5$. Despite these successes, there are still several unanswered questions regarding randomized smoothing based certified defenses. One of such questions is, why should Gaussian noise be used for randomized smoothing to certify $\ell_2$-normed robustness, and is Gaussian mechanism the best option? Another important question is regarding the generalizability of this method to other norms, especially the $\ell_\infty$ norm. If randomized smoothing can be used to certify $\ell_\infty$-normed robustness, what mechanism is the optimal choice?

To shed light on the above questions, we propose in this paper a unified and self-contained framework for randomized smoothing-based certified defenses. We look at the problem from a differential privacy's point of view and present two types of robustness in this framework. One is motivated by

| Mechanism | $\ell_2$-normed | | $\ell_\infty$-normed | |
| | $D_\infty$ Robustness | $D_{MR}$ Robustness | $D_\infty$ Robustness | $D_{MR}$ Robustness |
|---|---|---|---|---|
| Gaussian | unable to certify | near optimal $r$ scales in $O(1)$ | unable to certify | near optimal $r$ scales in $O(1/\sqrt{d\log d})$ |
| Exponential | not optimal | not optimal | optimal $r$ scales in $O(1/d)$ | not optimal |

Table 1: Summary of our framework

$\epsilon$-differential privacy ($\epsilon$-DP), which uses $\infty$-divergence to measure the distance between the probabilities of predictions on randomized natural samples and randomized adversarial samples and is therefore called $D_\infty$ robustness. The other is inspired by $\epsilon$-zero concentrated differential privacy ($\epsilon$-zCDP) that uses the Maximal Relative Rényi (MR) divergence as the probability distance measurement and is called $D_{MR}$ robustness. For both of them, we focus on certifying robustness in either $\ell_2$ or $\ell_\infty$ norm by randomized smoothing. Specifically, our contributions are five-fold:

1. We propose a unified and self-contained framework for certifying $D_\infty$ and/or $D_{MR}$ robustness in $\ell_2$ and $\ell_\infty$ norms by randomized smoothing.

2. In our framework, we demonstrate that the Gaussian mechanism is a near optimal choice for certifying $D_{MR}$ robustness in $\ell_2$ norm, and the robust radius is $O(1)$.

3. We also prove that an exponential mechanism is the optimal choice for certifying $D_\infty$ robustness in $\ell_\infty$ norm, but the robust radius is only $O(1/d)$, making it unscalable to high-dimensional data.

4. We show that the Gaussian mechanism is also a near optimal choice for certifying $D_{MR}$ robustness in $\ell_\infty$ norm, but the robust radius is $O(1/\sqrt{d\log d})$, making it also hardly scalable to high-dimensional data.

5. The largest robust $\ell_\infty$ radius that can be certified by randomized smoothing to achieve $D_{MR}$ robustness is upper bounded by $O(1/\sqrt{d})$.

Table 1 summarizes the (near) optimal mechanisms of our framework for certifying the $\ell_2$ and $\ell_\infty$-normed robustness.

## 2 RELATED WORK

There are three main approaches for certified defenses. The first approach formulates the task of adversarial verification as an optimization problem and solves it by relaxations (Dvijotham et al., 2018; Raghunathan et al., 2018; Wong & Kolter, 2018). The second approach uses different techniques, such as interval analysis and abstract interpretations, to maintain an outer approximation of the output at each layer through the network. (Mirman et al., 2018; Wang et al., 2018; Gowal et al., 2018). The third approach uses randomized smoothing to certify robustness, and is gaining popularity recently due to its strong scalability (Lecuyer et al., 2018; Li et al., 2018; Cohen et al., 2019) to large datasets and arbitrary networks. For this approach, Lecuyer et al. (2018) showed that randomized smoothing can certify the $\ell_2$ and $\ell_1$-normed robustness by using inequalities from differential privacy. Li et al. (2018) achieved a stronger guarantee on the $\ell_2$-normed robustness using tools from information theory. Cohen et al. (2019) further obtained a tight guarantee on the $\ell_2$-normed robustness using Gaussian noise. A remaining issue in all of these works is that they did not give answers to questions like why Gaussian noise is used to certify the $\ell_2$-normed robustness and what is the best mechanism to certify the $\ell_\infty$-normed robustness. To answer these questions, we present in this paper a new general framework to study randomized smoothing based certified defenses.

## 3 ROBUSTNESS MOTIVATED BY DIFFERENTIAL PRIVACY

In this section, we introduce our framework. Let $\mathbf{x}$ be a data sample and $y \in \mathcal{Y}$ be its label, where $\mathcal{Y}$ is the label set. We denote by $f(\cdot)$ a deterministic classifier with prediction $f(x)$ for any data sample $x$. If there exists an $\mathbf{x}'$ in a small $l_p$ ball centered at $\mathbf{x}$ and with $f(\mathbf{x}') \neq f(\mathbf{x})$, $\mathbf{x}'$ is viewed as an adversarial sample.

**Definition 1** (Randomized Classifier (Cohen et al., 2019)). *Given an input $x$, the prediction of a randomized classifier $g(\cdot)$ is defined as*

$$\operatorname*{argmax}_{c \in \mathcal{Y}} P(g(x) = c).$$

*Specifically, for a randomized smoothing classifier $g(x) = f(x + Z)$, where $Z$ is a random vector and $f(\cdot)$ is a deterministic classifier, the prediction of $x$ is the class of $c$ whose region $S \triangleq \{\tilde{x} \in R^d, f(\tilde{x}) = c\}$ has the largest probability measure in the distribution of $x + Z$ ($\tilde{x} \sim p(x + Z)$).*

Before introducing our framework, we first recall the definition of robustness for a deterministic classifier in (Diochnos et al., 2018).

**Definition 2** (Robustness (Diochnos et al., 2018)). *For a given classifier $f$, a sample $x$ and some norm $\| \cdot \|$. $f$ is $(r, \| \cdot \|)$-(error-region) robust on the sample $x$ if*

$$\forall x' \in \mathbb{B}(x, r), f(x) = f(x'), \tag{1}$$

*where $\mathbb{B}(x, r)$ is the ball centered at $x$ and with norm $\| \cdot \|$ and radius $r$.*

Note that in Definition 2, the classifier is assumed to be deterministic. To generalize the concept of robustness to randomized classifiers (see Definition 1), we define a relaxed version of the (error-region) robustness. Since $g(x)$ is a random value, instead of using equality, we measure the difference between $g(x)$ and $g(x')$ by a certain divergence. This leads us to the following definition, which is a basic concept in our framework that will be used throughout the paper.

**Definition 3** (Relaxed Robustness). *For a given (randomized) classifier $g(\cdot)$, a sample $x$ and some norm $\| \cdot \|$, the classifier $g$ is $(r, D, \| \cdot \|, \epsilon)$-(error-region) robust on $x$ if*

$$\forall x' \in \mathbb{B}(x, r), \max\{D(g(x), g(x')), D(g(x'), g(x))\} \le \epsilon. \tag{2}$$

*where $D$ is some divergence metric between two probability distributions. The $\max$ function is used to ensure that the measurement is symmetric.*

Compared with Definition 2, there are two additional terms in Definition 3: $\epsilon$ represents the "distance" or difference between the distributions of $g(x)$ and $g(x')$. When $\epsilon$ is small, we expect that the distributions of predictions on $x$ and $x'$, *i.e.*, $g(x)$ and $g(x')$, are almost the same, which is just a generalization of the equality in Definition 2. $D$ is some divergence measurement between two probability distributions. In this paper, we use two types of divergence, $\infty$-Divergence and Maximal Relative Rényi Divergence, to measure the distance between two probability distributions. Correspondingly, we have two types of robustness called $D_\infty$ and $D_{MR}$ robustness.

**Definition 4** ($\infty$-Divergence). *The $\infty$-Divergence $D_\infty$ of distributions $P$ and $Q$ is defined as*

$$D_\infty(P\|Q) = \sup_{x \in supp(Q)} \log \frac{P(x)}{Q(x)},$$

*where $supp(Q)$ is the support of the distribution $Q$.*

**Definition 5** (Maximal Relative Rényi Divergence). *The Maximal Relative Rényi Divergence $D_{MR}(P\|Q)$ of distributions $P$ and $Q$ is defined as*

$$D_{MR}(P\|Q) = \max_{\alpha \in (1, \infty)} \frac{D_\alpha(P\|Q)}{\alpha},$$

*where $D_\alpha(P\|Q)$ is the Rényi divergence between $P$ and $Q$, which is defined as*

$$D_\alpha(P\|Q) = \frac{1}{\alpha - 1} \log \mathbb{E}_{x \sim Q}(\frac{P(x)}{Q(x)})^\alpha.$$

**Definition 6** ($D_\infty$ Robustness). *A randomized smoothing mechanism $\mathcal{A}(\cdot)$ (including classifiers) is a $(r, D_\infty, \| \cdot \|, \epsilon)$-robust mechanism if*

$$\forall x' \in \mathbb{B}(x, r), \max\{D_\infty(\mathcal{A}(x), \mathcal{A}(x')), D(\mathcal{A}(x'), \mathcal{A}(x))\} \le \epsilon, \tag{3}$$

*where $\| \cdot \|$ is the norm of the ball $\mathbb{B}(x, r)$. If a randomized smoothing classifier $g(\cdot)$ satisfies Eq. (3), it is a $(r, D_\infty, \| \cdot \|, \epsilon)$-robust classifier or it certifies $D_\infty$ Robustness.*

$D_\infty$ Robustness is motivated by the notion of $\epsilon$-differential privacy ($\epsilon$-DP) (Dwork et al., 2006). To achieve $\epsilon$-DP for a randomized algorithm, we can use several mechanisms such as Laplacian mechanism or Exponential mechanism (see (Dwork et al., 2014) for details). However, it is known that adding Gaussian noise often does not lead to $\epsilon$-DP, but rather $(\epsilon, \delta)$-DP (Dwork et al., 2014) which has an additional parameter $\delta$ and thus is harder to be incorporated in our framework. To alleviate this issue, we employ Maximal Relative Rényi Divergence as the the probability distance measurement to define another type of robustness, namely $D_{MR}$ robustness.

**Definition 7** ($D_{MR}$ Robustness). *A randomized smoothing mechanism $\mathcal{A}(\cdot)$ is a $(r, D_{MR}, \|\cdot\|, \epsilon)$-robust mechanism if*

$$\forall x' \in \mathbb{B}(x, r), \max\{D_{MR}(\mathcal{A}(x), \mathcal{A}(x')), D_{MR}(\mathcal{A}(x'), \mathcal{A}(x))\} \le \epsilon. \tag{4}$$

*If a randomized smoothing classifier $g(\cdot)$ satisfies Eq. (4), it is a $(r, D_{MR}, \|\cdot\|, \epsilon)$-robust classifier or it certifies $D_{MR}$ Robustness.*

$D_{MR}$ Robustness is inspired by the notion of zero-Concentrated Differential Privacy (zCDP) (Bun & Steinke, 2016), whose connection to DP is shown in the following theorem.

**Theorem 8** ((Bun & Steinke, 2016)). *Let $P$ and $Q$ be two probability distributions satisfying the conditions of $D_\infty(P\|Q) \le \epsilon$ and $D_\infty(Q\|P) \le \epsilon$. Then, $D_{MR}(P\|Q) \le \frac{1}{2}\epsilon^2$.*

Theorem 8 indicates that $D_{MR}$-robustness is a relaxed version of $D_\infty$-robustness.

**Remark** (Connections between $D_\infty$ & $D_{MR}$ Robustness and Standard Definitions). *Although $D_\infty$ & $D_{MR}$ Robustness are seemingly new concepts defined in this paper, they actually have several connections with the existing frameworks Lecuyer et al. (2018) and Cohen et al. (2019). Specifically, as long as $D_\infty$ robustness is certified, the expected output stability bound in Lecuyer et al. (2018) will be guaranteed with $\delta' = 0$. And if $D_{MR}$ robustness is certified, the expected output stability bound in Lecuyer et al. (2018) will be guaranteed with $\epsilon' = (c+1)\sqrt{\epsilon}$ and $\delta' = \exp(-\frac{c^2}{4})$, according to Theorem 10. Besides, the "scale" of the robust radius certified by our framework is similar the "scale" of the robust radius in Cohen et al. (2019), according to Corollary 11.*

**Theorem 9** (Postprocessing Property). *Let $g(x) = f(\mathcal{A}(x))$ be a randomized classifier, where $f(\cdot)$ is any deterministic function (classifier). $g(\cdot)$ is $(r, D, \|\cdot\|, \epsilon)$-robust if $\mathcal{A}(\cdot)$ is $(r, D, \|\cdot\|, \epsilon)$-robust (where $D$ includes $D_\infty$ and $D_{MR}$).*

The above theorem is derived from the post-processing properties of DP and zCDP. A detailed proof (explanation) is given in Appendix B. **This property allows us to concentrate only on the randomized smoothing mechanism $\mathcal{A}$ without needing to consider the specific form of the deterministic function (classifier) $f(\cdot)$.** Next, we consider the cases of certifying $D_\infty$ or $D_{MR}$ robustness using $\ell_2$ and $\ell_\infty$-norm.

## 3.1 Certifying $\ell_2$-normed Robustness

The following theorem shows that randomized smoothing by the Gaussian mechanism is $(r, D_{MR}, \|\cdot\|, \epsilon)$-robust.

**Theorem 10.** *Let $f$ be any classifier and $g(x) = f(x+z)$ be its corresponding randomized classifier for samples $x \in \mathbb{R}^d$, where $z \sim \mathcal{N}(0, \sigma^2 I_d)$. Then, $g(\cdot)$ is $(r, D_{MR}, \|\cdot\|_2, \frac{r^2}{2\sigma^2})$-robust on any $x$. Moreover, let $\epsilon$ denote $\frac{r^2}{2\sigma^2}$. Then, for any $\lambda > 0$ and any measurable set $S \ne \emptyset$, the following holds with probability at least $1 - \exp(-\frac{\lambda^2}{4\epsilon})$,*

$$\log \frac{P(g(x) \in S)}{P(g(x') \in S)} \le \lambda + \sqrt{\epsilon}. \tag{5}$$

*That is, when $\lambda = c\sqrt{\epsilon}$, $\log \frac{P(g(x) \in S)}{P(g(x') \in S)} \le (c+1)\sqrt{\epsilon}$ with probability $1 - \exp(-\frac{c^2}{4})$. In practice, $c = 3$ is enough to achieve a high probability.*

**Corollary 11.** *Adding Gaussian noise $z \in \mathcal{N}(0, \sigma^2 I_d)$ can defend any $x' \in \mathbb{B}(x, r = \sqrt{2\epsilon}\sigma)$ that satisfies the condition of $D_{MR}(g(x)\|g(x')) \le \epsilon$ with probability at least $1 - \exp(-\frac{c^2}{4})$. Furthermore, $\sqrt{\epsilon}$ can be calculated (bounded) by $(\log p_a - \log p_b)/2(1+c)$ or $(\log p_a/(1-p_a))/2(1+c)$ (binary case), where $p_a$ and $p_b$ are respectively the probabilities of the randomized classifier $g(\cdot)$ returning the most probable class $c_a$ and the runner-up class $c_b$ on input $x$.*

***Detailed proofs for Theorem 10, Corollary 11, and all the following theorems are provided in Appendix B.*** From Theorem 9, we can see that for classifiers like $g(x) = f(x + z)$, we only need to prove that the randomized mechanism $\mathcal{A}(x) = x + z(z \sim \mathcal{N}(0, \sigma^2 I_d))$ is $(r, D_{MR}, \|\cdot\|_2, \frac{r^2}{2\sigma^2})$-robust. Also, the connection between $\epsilon$ and $p_a, p_b$ can be derived for all $\epsilon$ or $\sqrt{\epsilon}$ (in the certified radii) as in Corollary 11. Note that a similar theorem has also been proved by Cohen et al. (2019). But there are some major differences between our framework and theirs (Cohen et al., 2019). Specifically, our framework certifies the robustness with a probability of failure, and the certified radius $r$ depends on $c$ that controls the probability of failure. A smaller $c$ yields a larger $r$ compared to those in Cohen et al. (2019), and vice versa. Moreover, in our framework, we show that the Gaussian mechanism is a near optimal option, by providing a lower bound below for all possible noises that can certify the $\ell_2$-normed $D_{MR}$ robustness.

Next, we consider the following unanswered question (*i.e.*, the first question). Since there are infinite ways of sampling $z$, a natural problem is to determine whether Gaussian mechanism is the optimal option to certify the $\ell_2$-normed $D_{MR}$ robustness. To answer this question, we first give ***a lower bound*** on the magnitude of the noise $z$ added in the *randomized smoothing mechanism $\mathcal{A}(x) = x + z$* to ensure that $\mathcal{A}(x)$, as well as $f(\mathcal{A}(x))$, is $(r, D_{MR}, \|\cdot\|_2, \epsilon)$-robust. If the magnitude of Gaussian noise is close to the lower bound, then Gaussian mechanism is considered as "near optimal".

**Theorem 12** (Lower Bound of the Noise). *For any $\epsilon \leq O(1)$, if there is a $(2r, D_{MR}, \|\cdot\|_2, \frac{\epsilon}{2})$-robust randomized smoothing mechanism $\mathcal{A}(x) = x + z : [0, \frac{r}{\sqrt{d}}]^d \mapsto [0, \frac{r}{\sqrt{d}}]^d$ such that for all $x \in [0, \frac{r}{\sqrt{d}}]^d$,*

$$\mathbb{E}[\|z\|_\infty] = \mathbb{E}_{\mathcal{A}}\|\mathcal{A}(x) - x\|_\infty \leq \alpha,$$

*for some $\alpha \leq O(1)$, then it must be true that $\alpha \geq \Omega(\frac{r}{\sqrt{\epsilon}})$. In another word, $\Omega(\frac{r}{\sqrt{\epsilon}})$ is the lower bound of the expected $\ell_\infty$ norm of the random noise.*

Theorem 12 indicates that the expected $\ell_\infty$ norm of the added random noise should be at least $\Omega(\frac{r}{\sqrt{\epsilon}})$ to guarantee $(r, D_{MR}, \|\cdot\|_2, \epsilon)$-robustness. For Gaussian mechanism, the expected $\ell_\infty$ norm is $O(\sigma\sqrt{\log d})$ ((Orabona & Pál, 2015)), which is $O(\frac{r}{\sqrt{\epsilon}}\sqrt{\log d})$ according to Corollary 11. This means that Gaussian mechanism is near optimal (i.e., up to an $O(\sqrt{\log d})$ factor) here. Equivalently, if we fix the magnitude of the expected $\ell_\infty$-norm of the added noise as $\alpha$, the largest radius $r$ that can be certified by any $(r, D_{MR}, \|\cdot\|_2, \epsilon)$-robust randomized smoothing mechanisms is upper bounded by $O(\alpha\sqrt{\epsilon})$, which is also close to the robust radius guaranteed by Gaussian mechanism (up to an $O(\sqrt{\log d})$ factor).

## 3.2 CERTIFYING $\ell_\infty$-NORMED ROBUSTNESS

Previous work on the randomized smoothing-based certified defenses (Cohen et al., 2019; Li et al., 2018) mainly uses Gaussian noise to certify the $\ell_2$-normed robustness. Thus, another natural question (*i.e.*, the second question) is to determine whether randomized smoothing can use some mechanism to certify the $\ell_\infty$-normed robustness. In this section, we consider this question using our general framework.

Before extending our result to the $\ell_\infty$-normed case, we first recall the $\ell_2$-normed case and investigate the form of the density function of Gaussian noise: $p(\boldsymbol{z}) \propto \exp(-\frac{\|\boldsymbol{z}\|_2^2}{\sigma^2})$. Based on this, we conjecture that, to certify $\ell_\infty$-normed robustness, we can sample the noise using an exponential mechanism:

$$p(\boldsymbol{z}) \propto \exp\left(-\frac{\|\boldsymbol{z}\|_\infty}{\sigma}\right). \tag{6}$$

We show in the following theorem that randomized smoothing by (6) certifies $(r, D_{MR}, \|\cdot\|_\infty, \cdot)$-robustness, which could be considered as an extension of the $\ell_2$-normed case. Moreover, we can prove that it is $(r, D_\infty, \|\cdot\|_\infty, \cdot)$-robust. However, the certified radius $r$ is $O(1/d)$, which implies that it is unscalable to high-dimensional data.

**Theorem 13.** *Let $f$ be any classifier and $g(x) = f(x+z)$ be its corresponding randomized classifier for sample $x \in \mathbb{R}^d$, where the noise $z \sim p(z)$ in (6). Then, $g(\cdot)$ is $(r, D_{MR}, \|\cdot\|_\infty, \frac{r^2}{2\sigma^2})$-robust. Moreover, it is $(r, D_\infty, \|\cdot\|_\infty, \frac{r}{\sigma})$-robust.*

**Remark 14.** *Due to the high dimensionality of samples in real world applications, directly sampling $z \sim p(z)$ by the Markov Chain Monte Carlo (MCMC) algorithm requires a large number of random-walks that can incur high computational cost. To alleviate this issue, we adopt an efficient sampling method from (Steinke & Ullman, 2015) that first samples $R$ from $Gamma(d+1, \sigma)$ and then samples $\mathbf{z}$ from $[-R, R]^d$ uniformly. The complexity of this sampling algorithm is only $O(d)$.*

Comparing Theorems 10 and 13, we can see that randomized smoothing via (6) can certify a region that has (almost) the same radius as that of Gaussian distribution in the $\ell_2$-normed case, due to similarity in their density functions and the robustness guarantees. In the following theorem we show that the magnitude of the noise added by (6) is much larger than that of Gaussian distribution in the $\ell_2$-normed case.

**Theorem 15.** *For the distribution that can guarantee Theorem 13, the following theorem holds*

$$\mathbb{E}_z[\|z\|_\infty] = d\sigma. \tag{7}$$

Note that compared with the Gaussian noise added in Theorem 10 which satisfies the condition of $\mathbb{E}_z[\|z\|_\infty] = O(\sigma\sqrt{\log d})$, the expected $\ell_\infty$-norm of the distribution in (6) is proportional to the dimensionality $d$ of the data, which is quite large. This means that for any image data, at least one pixel will be perturbed by the magnitude of $d\sigma$, which will completely ruin the accuracy of the classification network. However, if we want the noise to have a magnitude of $O(1)$, $\sigma$ needs to be $O(1/d)$, and so does the robust radius.

Theorem 15 is a somewhat negative result for randomized smoothing using distribution (6) to certify the $\ell_\infty$-normed robustness. Thus, an immediate question is whether exponential mechanism is the right choice to certify the $\ell_\infty$-normed robustness. The following theorem shows that for any $(r, D_\infty, \|\cdot\|_\infty, \frac{r}{\sigma})$-robust randomized smoothing mechanism, the expected $\ell_\infty$-norm of the added noise is lower bounded by $\Omega(d\sigma)$. Thus, combining the following theorem with Theorem 15, we can conclude that the exponential mechanism is actually an optimal choice to certify $D_\infty$ robustness.

**Theorem 16.** *For any $(2r, D_\infty, \|\cdot\|_\infty, \frac{\epsilon}{2})$-robust mechanism $\mathcal{A}(x) = x + z : [0, r]^d \mapsto [0, r]^d$ such that*

$$\mathbb{E}[\|z\|_\infty] = \mathbb{E}_\mathcal{A}\|\mathcal{A}(x) - x\|_\infty \le \alpha, \forall x \in [0, r]^d,$$

*it must be true that $\alpha \ge \Omega(\frac{rd}{\epsilon})$.*

From Theorem 16 we can see that, for any $(\cdot, D_\infty, \|\cdot\|_\infty, \frac{\epsilon}{2})$-robust randomized smoothing mechanism, if we fix the expectation of the $\ell_\infty$-norm of the added noise in the exponential mechanism as $\alpha$, the largest $\ell_\infty$ radius that can be certified is upper bounded by $O(\alpha\epsilon/d)$. Compared with the $\ell_2$-normed case in Theorem 11, we can see that there is an additional factor of $O(1/d)$, which makes it unscalable to high-dimensional data. Equivalently, if we want the same radius to be certified as in the Theorem 10, the expected $\ell_\infty$-norm of the added noise needs to be at least $\Omega(\frac{rd}{\epsilon})$, which will be too large for any image data.

The less than ideal lower bound in Theorem 16 is for $D_\infty$-robustness. Since $D_{MR}$-robustness is more relaxed than $D_\infty$-robustness, a natural question is thus to determine whether the lower bound can be improved by switching to $D_{MR}$-robustness. Unfortunately, the following theorem shows that a similar phenomenon still holds for $D_{MR}$-robustness.

**Theorem 17.** *For any $(2r, D_{MR}, \|\cdot\|_\infty, \frac{\epsilon}{2})$-robust mechanism $\mathcal{A}(x) = x + z : [0, r]^d \mapsto [0, r]^d$ such that*

$$\mathbb{E}[\|z\|_\infty] = \mathbb{E}_\mathcal{A}\|\mathcal{A}(x) - x\|_\infty \le \alpha, \forall x \in [0, r]^d,$$

*it must be true that $\alpha \ge \Omega(\frac{r\sqrt{d}}{\sqrt{\epsilon}})$.*

From Theorems 17 and 15 we can see that in the definition of $(2r, D_{MR}, \|\cdot\|_\infty)$-robustness, adding noise according to (6) is not near optimal. The following theorem shows that in this case, Gaussian mechanism is actually a near optimal choice.

**Theorem 18.** *Let $r, \epsilon > 0$ be some fixed number and $\mathcal{A}(x) = x + z$ with $z \sim \mathcal{N}(0, \frac{dr^2}{2\epsilon})$. Then, $\mathcal{A}(\cdot)$ is $(r, D_{MR}, \|\cdot\|_\infty, \epsilon)$-robust. $\mathbb{E}[\|z\|_\infty] = \mathbb{E}_\mathcal{A}\|\mathcal{A}(x) - x\|_\infty$ is upper bounded by $O(\frac{r\sqrt{d\log d}}{\sqrt{\epsilon}})$.*

From Theorem 17 and 18, we can conclude that for all randomized smoothing mechanisms that are $(\cdot, 0, D_{MR}, \|\cdot\|_\infty, \frac{\epsilon}{2})$-robust, if the expected $\ell_\infty$-norm of the added noise is fixed to be $\alpha$, the

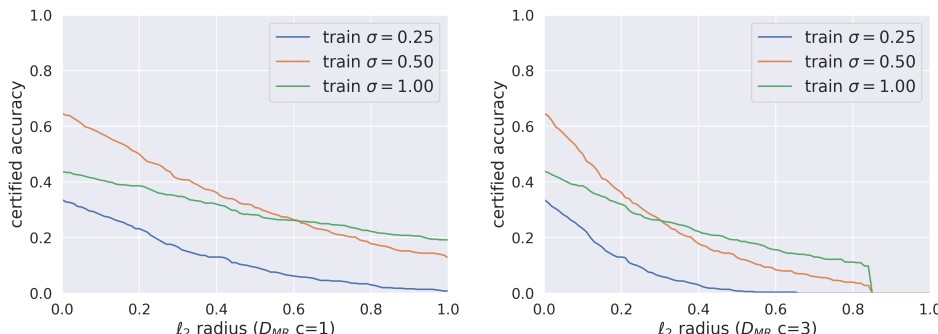

Figure 1: Certifying $D_{MR}$ robustness in $\ell_2$ norm on CIFAR-10: vary the Gaussian noise used in the training process and fix the $\sigma$ of the Gaussian mechanism as $\sigma = 0.5$. $c = 1$ (left) and $c = 3$ (right)

largest radius that can be certified is upper bounded by $O(\frac{\sqrt{\epsilon}\alpha}{\sqrt{d}})$, and the largest radius that can be certified by Gaussian mechanism is $O(1/\sqrt{d \log d})$ (and $\sigma$ is $\Omega(\frac{\alpha}{\sqrt{\log d}})$). If $\alpha$ and $\epsilon$ are both set to be $O(1)$, the largest radius that can be certified using Gaussian mechanism to achieve $D_{MR}$-robustness is greater than the largest radius that can be certified to achieve $D_\infty$-robustness by at least a factor of $O(\sqrt{d/\log d})$. This is reasonable since the definition of $D_{MR}$-robustness is more relaxed. Obviously, there is some **trade-off** between the rigorousness of the notion of robustness and the largest certified robust radius, *i.e.,* when the robustness is relaxed, the largest certified radius increases. We will investigate this trade-off more in the future research.

## 4 EXPERIMENTS

### 4.1 DATASETS AND MODELS

The performance of our framework is verified on two widely-used datasets, *i.e.,* CIFAR10 and ImageNet[*]. Following Cohen et al. (2019), we use a 110-layer residual network and the classical ResNet-50 as the base models for CIFAR10 and ImageNet respectively. Note that it may be difficult for the models to classify noisy images without seeing any noisy samples in the training stage. Thus, we train all the models by adding appropriate Gaussian noise on the training images. The certified accuracy for radius $R$ is defined as the fraction of the test set whose certified radii are larger than $R$ [†]. The value of $\epsilon$ in all our derived certified radii can be calculated by $p_a$ (or $p_a$ and $p_b$) as shown in the proof of Corollary 11. It is also worth noting that we do not compare our results with (Cohen et al., 2019) in the experiments because our framework and (Cohen et al., 2019) endow robustness with different definitions. *Moreover, our work does not aim at improving the tightness of the guarantee on the $\ell_2$-normed robustness but aims at presenting a general and self-contained framework to study some remaining issues, such as the optimality of the Gaussian mechanism, and the specific mechanisms to certify the $\ell_\infty$-normed robustness.*

### 4.2 EMPIRICAL RESULTS

**Certifying the $\ell_2$-normed Robustness** To certify the $\ell_2$-normed Robustness, as we explained in previous section, Gaussian mechanism is a near optimal option. Thus, we mainly evaluate the performance of Gaussian mechanism in our framework. We first fix the value of $\sigma$ in Gaussian mechanism and show the certified accuracy of the classifiers trained by varied Gaussian noises in Figure 1. As shown in Figure 1, using $\sigma = 0.50$ Gaussian noise to train the classifier is a good setting here. So in Figure 2, we evaluate the Gaussian mechanism with different $\sigma$ values on the classifier trained by $\sigma = 0.50$ Gaussian noise. Overall, on CIFAR-10, our framework can certify approximately 20% accuracy under $\ell_2 = 1.0$ perturbation[‡]. We also show the results on ImageNet by Figures 4 and 5 in Appendix C.

---

[*]Pixel value range is $[0.0, 1.0]$

[†]For more details, please refer to (Cohen et al., 2019)

[‡]On CIFAR-10, $\ell_2 = 1.0$ perturbation allows $4/255$ perturbation on every pixel

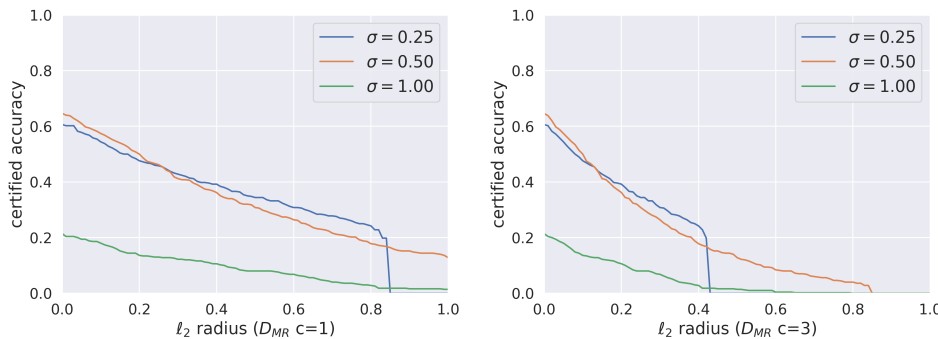

Figure 2: Certifying $D_{MR}$ robustness in $\ell_2$ norm on CIFAR-10: vary the $\sigma$ in the Gaussian mechanism and fix $\sigma$ of the training noise as $\sigma = 0.50$. $c = 1$ (left) and $c = 3$ (right)

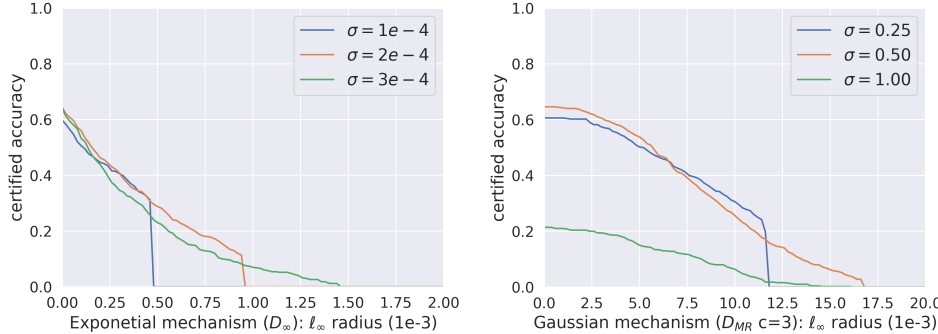

Figure 3: Certifying $D_\infty$ robustness and $D_{MR}$ robustness in $\ell_\infty$ norm on CIFAR-10: vary the $\sigma$ in the Exponential mechanism (left) vary the $\sigma$ in the Gaussian mechanism (right). The classifier is trained with $\sigma = 0.50$ Gaussian noise.

**Certifying the $\ell_\infty$-normed Robustness** To certify the $\ell_\infty$-normed robustness, we evaluate the performance of the Exponential mechanism in the definition of $D_\infty$-robustness and the Gaussian mechanism in the definition of $D_{MR}$-robustness. As shown in Figure 3, the $\ell_\infty$ radii that can be certified by Gaussian mechanism are about $10 \sim 20$ times (*i.e.,* $O(\sqrt{d/\log d})$ with $d = 3072$ as shown in our theories) larger than the $\ell_\infty$ radii certified by the exponential mechanism. On ImageNet, as shown in Figure 6 in Appendix C, the robust radii are less than $1/255$ (due to scaling in $O(1/d)$ or $O(1/\sqrt{d \log d})$), indicating that certifying the $\ell_\infty$-normed robustness by randomized smoothing may not be applicable to high-dimensional data.

## 5 CONCLUSION

In this paper, we present a general framework for certifying two types of robustness ($D_\infty$ and $D_{MR}$-robustness) in the $\ell_2$ and $\ell_\infty$ norms by randomized smoothing. Under our framework, we first give the answers to the remaining questions in the previous studies on randomized smoothing-based certifiable defenses, *i.e.,* the optimality of Gaussian mechanism and the possibility to certify the $\ell_\infty$-normed robustness. Specifically, we demonstrate that (i) Gaussian mechanism is a near optimal option to certify $D_{MR}$-robustness in $\ell_2$ norm by giving a lower bound on all $D_{MR}$-robust mechanisms, with certified radii scaling in $O(1)$; (ii) an exponential mechanism is the optimal choice for certifying $D_\infty$-robustness in $\ell_\infty$ norm, with certified radii scaling in $O(1/d)$; (iii) Gaussian mechanism is a near optimal option to certify $D_{MR}$-robustness in $\ell_\infty$ norm, with certified radii scaling in $O(1/\sqrt{d \log d})$; (iv) the largest $\ell_\infty$ radius that can be certified by randomized smoothing in our framework is upper bounded by $O(1/\sqrt{d})$, indicating that randomized smoothing may not be scalable to high-dimensional data in terms of certifying the $\ell_\infty$-normed robustness.

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

## A  DIFFERENTIAL PRIVACY BACKGROUND

In this section, we briefly introduce the concepts of differential privacy used in this paper.

**Definition 19** (Differential Privacy (DP) (Dwork et al., 2006))**.** *Given a data universe $\mathcal{X}$, we say that two datasets $D, D' \subseteq \mathcal{X}$ are neighbors if they differ by only one entry, which is denoted by $D \sim D'$. A randomized algorithm $\mathcal{A}$ is $\epsilon$-differentially private (DP) if for all neighboring datasets $D, D'$ the following holds*

$$D_\infty(\mathcal{A}(D)\|\mathcal{A}(D')) \leq \epsilon.$$

Intuitively, DP ensures that an adversary cannot infer whether or not a participant (data sample) is participating in dataset $D$ due to the fact that the distribution of $\mathcal{A}(D)$ is almost the same as that of $\mathcal{A}(D')$, which means that DP-mechanisms are robust to 1-sample change. Now consider the case where D is some 1-size dataset (*i.e.,* one data sample). Then, DP ensures that the distribution of $\mathcal{A}(D)$ and $\mathcal{A}(D')$ are almost the same, where $D'$ is just any other data sample. Inspired by notion of DP, we define $D_\infty$ robustness in Definition 6.

**Definition 20** (Zero-Concentrated Differential Privacy (zCDP))**.** *A randomized mechanism $\mathcal{A}$ is called $\epsilon$-zCDP, if for all $D \sim D'$*

$$\max\{D_{MR}(\mathcal{A}(D)\|\mathcal{A}(D')), D_{MR}(\mathcal{A}(D')\|\mathcal{A}(D))\} \leq \epsilon. \tag{8}$$

zCDP is a relaxed version of DP according to Theorem 8. Motivated by zCDP, we define $D_{MR}$ robustness in Definition 7.

## B  OMITTED PROOFS

*Proof of Theorem 9.* This theorem can be easily proved by the following lemma,

**Lemma 21** ((Bun & Steinke, 2016))**.** *Let $P$ and $Q$ be two distributions on $\Omega$ and let $f : \Omega \mapsto \Theta$ be a deterministic function. Let $f(P)$ and $f(Q)$ denote the distributions on $\Theta$ induced by applying $f$ to $P$ and $Q$ respectively. Then we have*

$$D_\alpha(f(P)\|f(Q)) \leq D_\alpha(P\|Q).$$

Similar post-processing property also holds when $\alpha = \infty$ (Dwork et al., 2006). Therefore, if $\mathcal{A}(\cdot)$ satisfies Definition 6 or 7, then $f(\mathcal{A}(\cdot))$ will satisfy Definition 6 or 7 for any deterministic function (classifier) $f(\cdot)$. □

*Proof of Theorem 10.* By Theorem 9, we only need to show that the randomized smoothing mechanism $\mathcal{A}(x) = x + z$ is $(r, D_{MR}, \|\cdot\|_2)$-robust, which can be proved by the following lemma.

**Lemma 22** ((Bun & Steinke, 2016))**.** *Let $x, x' \in \mathbb{R}^d$, and $\alpha \in [1, \infty)$. Then*

$$D_\alpha(\mathcal{N}(x, \sigma^2 I_d)\|\mathcal{N}(x', \sigma^2 I_d)) = \frac{\alpha\|x - x'\|_2^2}{2\sigma^2}.$$

*Thus for all $x' \in \mathbb{B}(x, r)$, we have $D_{MR}(\mathcal{A}(x)\|\mathcal{A}(x')) \leq \frac{r^2}{2\sigma^2}$.*

Next we prove (5). To prove this inequality, we first define the loss random variable.

**Definition 23** ((Bun & Steinke, 2016))**.** *Let $Y$ and $Y'$ be random variables on $\Omega$. We define the loss random variable between $Y$ and $Y'$, denoted by $Z = Loss(Y\|Y')$, as follows: Define a function $F : \Omega \mapsto \mathbb{R}$ by $F(y) = \log \frac{\mathbb{P}[Y=y]}{\mathbb{P}[Y'=y]}$. Then $Z$ is distributed according to $F(Y)$.*

By this we can write $Z = \text{Loss}(g(x)\|g(x'))$ and rewrite $D_{MR}(g(x)\|g(x'))$ as

$$\forall \alpha \in (1, \infty], \mathbb{E}[e^{(\alpha-1)Z}] \leq e^{(\alpha-1)\frac{r^2}{2\sigma^2}\alpha}.$$

This implies that $Z$ is sub-Gaussian. By using the tail-bound of sub-Gaussian (Vershynin, 2018), we have

$$\mathbb{P}[Z > \lambda + \epsilon] \leq \exp(-\frac{\lambda^2}{4\epsilon}), \tag{9}$$

where $\frac{r^2}{2\sigma^2} = \epsilon$. □

*Proof of Corollary 11.* Since we fix $\epsilon = \frac{r^2}{2\sigma^2}$, the certified radius is $r = \sqrt{2\epsilon}\sigma$. Now we prove the upper bound of $\sqrt{\epsilon}$ for a classifier $\mathbf{g}(\cdot)$. Given Theorem 10, we should have

$$\log \frac{P(g(x) = c_a)}{P(g(x') = c_a)} \leq (c+1)\sqrt{\epsilon},$$

and

$$\log \frac{P(g(x') = c_b)}{P(g(x) = c_b)} \leq (c+1)\sqrt{\epsilon},$$

since $x$ is also in $\mathbb{B}(x', r)$. Then we have $\log \frac{P(g(x')=c_a)}{P(g(x')=c_b)} \geq \log \frac{P(g(x)=c_a)}{P(g(x)=c_b)} - 2(c+1)\sqrt{\epsilon}$. According to Definition 1, as long as $\log \frac{P(g(x')=c_a)}{P(g(x')=c_b)} > 0$, $g(\cdot)$ can correctly classify $x'$. Thus, as long as $\log \frac{P(g(x)=c_a)}{P(g(x)=c_b)} - 2(c+1)\sqrt{\epsilon} > 0$ (*i.e.,* $\sqrt{\epsilon} < (\log p_a - \log p_b)/2(1+c)$), $g(\cdot)$ classifies $x'$ as $c_a$. □

*Proof of Theorem 12.* Let $\{x_1, x_2, \cdots, x_{2^d}\} = \{0, \frac{r}{\sqrt{d}}\}^d$. For each $x_i$, we use the same adversarial example $x' = 0$ to derive the lower bound. Since $\mathcal{A}$ is $(2r, D_{MR}, \|\cdot\|_2, \frac{\epsilon}{2})$-robust, we have for all $x_i, x_j, i, j \in [2^d]$,

$$\max\{D_{MR}(\mathcal{A}(x_i)\|\mathcal{A}(x_j)), D_{MR}(\mathcal{A}(x_j)\|\mathcal{A}(x_i))\} \leq 2 \cdot \frac{\epsilon}{2} = \epsilon.$$

*That is $\mathcal{A}$ is $\epsilon$-zCDP on the dataset $\mathcal{X} = \{0, \frac{r}{\sqrt{d}}\}^d$.* Next we will prove the lower bound for all $\epsilon$-zCDP mechanisms.

We first consider the case where $r = \sqrt{d}$, and then generalize it to any $r$. Before that we will first prove the lower bound of one-way marginal (*i.e.,* mean estimation) under $\epsilon$-zCDP. For an $n$-size dataset $X \in \mathbb{R}^{n \times d}$, the one-way marginal is just $h(D) = \frac{1}{n}\sum_{i=1}^{n} X_i$, where $X_i$ is the $i$-th row of $X$. Specifically, when $n = 1$, one-way marginal is just the data point itself. We show the following theorem,

**Theorem 24.** *If there exists an $\epsilon$-zCDP mechanism $\mathcal{A} : \{0,1\}^d \mapsto [0,1]^d$ such that for all $x \in \{0,1\}^d$*

$$\mathbb{E}\|\mathcal{A}(x) - x\|_\infty \leq \alpha, \tag{10}$$

*then $1 \geq \Omega(\sqrt{\frac{d}{\epsilon\alpha^2}})$.*

*Proof of Theorem 24.* To prove this theorem, our idea is to first use the connection between $\epsilon$-zCDP and $(\epsilon, \delta)$-DP.

**Lemma 25** (Prop.1.3 in Bun & Steinke (2016)). *If $\mathcal{A}$ is $\epsilon$-zCDP, then it is $(\epsilon + 2\sqrt{\epsilon \log \frac{1}{\delta}}, \delta)$-differentially private.*

Bun et al. (2018) first give the optimal rate of one-way marginal estimation which is improved by Steinke & Ullman (2016).

**Lemma 26** (Theorem 1.1 in Steinke & Ullman (2016)). *For every $\epsilon \leq O(1)$, every $2^{-\Omega(n)} \leq \delta \leq \frac{1}{n^{1+\Omega(1)}}$ and every $\alpha \leq \frac{1}{10}$, if $\mathcal{A} : (\{0,1\}^d)^n \mapsto [0,1]^d$ is $(\epsilon, \delta)$-DP and $\mathbb{E}[\|\mathcal{A}(D) - h(D)\|_\infty] \leq \alpha$, then*

$$n \geq \Omega(\frac{\sqrt{d \log \frac{1}{\delta}}}{\epsilon\alpha}). \tag{11}$$

Setting $n = 1, \epsilon = \epsilon + 2\sqrt{\epsilon \log \frac{1}{\delta}}$ in Lemma 26, we can see that if $\mathbb{E}[\|\mathcal{A}(x) - x\|_\infty] \leq \alpha$ then $1 \geq \Omega(\frac{\sqrt{d \log \frac{1}{\delta}}}{(\epsilon + 2\sqrt{\epsilon \log \frac{1}{\delta}})\alpha}) \geq \Omega(\frac{\sqrt{d}}{\sqrt{\alpha^2\epsilon}})$, where the last inequality is due to the fact that $\frac{\sqrt{\log \frac{1}{\delta}}}{\epsilon + 2\sqrt{\epsilon \log \frac{1}{\delta}}} \geq \Omega(\frac{1}{\sqrt{\epsilon}})$. □

Now we come back to the proof for any $r$. If $\mathcal{A} : \{0, \frac{r}{\sqrt{d}}\}^d \mapsto [0, \frac{r}{\sqrt{d}}]^d$ is $\epsilon$-zCDP, where $\mathbb{E}_{\mathcal{A}} \|\mathcal{A}(x_i) - x_i\|_\infty \leq \alpha$, then we have $\mathbb{E}_{\mathcal{A}} \|\frac{\sqrt{d}}{r} \mathcal{A}(x_i) - \frac{\sqrt{d}}{r} x_i\|_\infty \leq \frac{\sqrt{d}}{r} \alpha$. Thus, $\frac{\sqrt{d}}{r} \mathcal{A}$ is an $\epsilon$-zCDP mechanism on $\{0, 1\}^d \mapsto [0, 1]^d$. By Theorem 24 with $\alpha = \frac{\sqrt{d}}{r} \alpha \leq O(1)$, we have

$$1 \geq \Omega(\frac{r}{\sqrt{\epsilon \alpha^2}}), \quad \text{i.e., } \alpha \geq \Omega(\frac{r}{\sqrt{\epsilon}}). \tag{12}$$

$\square$

*Proof of Theorem 13.* We will first prove that $\mathcal{A}(x) = x + Z$ is $(r, D_\infty, \|\cdot\|_\infty, \frac{r}{\sigma})$-robust. Then by Theorems 8 and 9, we can easily show that $g(\cdot)$ is $(r, D_{MR}, \|\cdot\|_\infty, \frac{r^2}{2\sigma^2})$-robust. Consider $x, x', \|x' - x\|_\infty \leq r$. Then, for any $y$ we have

$$\frac{p(y - x)}{p(y - x')} = \frac{\exp(-\frac{\|y - x\|_\infty}{\sigma})}{\exp(-\frac{\|y - x'\|_\infty}{\sigma})} \leq \exp(\frac{\|y - x'\|_\infty - \|y - x\|_\infty}{\sigma}) \leq \exp(\frac{\|x' - x\|_\infty}{\sigma}) \leq \exp(\frac{r}{\sigma}).$$

Thus, for any subset $S$ we have

$$\log \frac{\mathcal{A}(x) \in S}{\mathcal{A}(x') \in S} = \log \frac{\int_S p(\mathbf{z} - x) d\mathbf{z}}{\int_S p(\mathbf{z} - x') d\mathbf{z}} \leq \frac{r}{\sigma}.$$

$\square$

*Proof of Theorem 15.* Define the distribution $D$ on $[0, \infty)$ to be $Z \sim D$, meaning $Z = \|z\|_\infty$ for $z \sim p(z)$, where $p(z)$ is in (6). The probability density function of $D$ is given by

$$p_D(z) \propto z^{d-1} \exp(-\frac{z}{\sigma}),$$

which is obtained by integrating the probability density function (6) over the infinity ball of radius $z$ with surface area $d2^d z^{d-1} \propto z^{d-1}$. $p_D$ is the Gamma distribution with shape $d$ and mean $\sigma$, and thus $\mathbb{E}[z] = d\sigma$. $\square$

*Proof of Theorem 16.* Let $\mathcal{X} = \{x_1, x_2, \cdots, x_{2^d}\} = \{0, r\}^d$ be the set of samples. Since $\mathcal{A}$ is $(2r, \|\cdots\|_\infty)$-robust and $\|x_i - x_j\|_\infty \leq 2r$, we know that

$$\max\{D_\infty(\mathcal{A}(x_i) \| \mathcal{A}(x_j)), D_\infty(\mathcal{A}(x_j) \| \mathcal{A}(x_i))\} \leq \epsilon.$$

Thus, $\mathcal{A} : \mathbb{R}^d \mapsto \mathbb{R}^d$ is $\epsilon$-DP on $\mathcal{X}$. Similar to the proof for Theorem 12, we can reduce our problem to studying the lower bound of one-way marginal for 1-size data problem in the $\epsilon$-DP model. Now we first consider the case of $r = 1$. We have the following lemma which is given by Hardt & Talwar (2010).

**Lemma 27** (Theorem 1.1 in (Hardt & Talwar, 2010)). *If there exists an $\epsilon$-DP mechanism $\mathcal{A} : \{0, 1\}^d \mapsto [0, 1]^d$ satisfying the following inequality for all $x \in \{0, 1\}^d$*

$$\mathbb{E}\|\mathcal{A}(x) - x\|_\infty \leq \alpha, \tag{13}$$

*then $1 \geq \Omega(\frac{d}{\epsilon \alpha})$.*

Now we consider any $\epsilon$-DP mechanism $\mathcal{A} : \{0, r\}^d \mapsto [0, r]^d$. If

$$\mathbb{E}[\|A(x) - x\|_\infty] \leq \alpha,$$

then $\mathbb{E}[\|\frac{1}{r} A(x) - \frac{1}{r} x\|_\infty] \leq \frac{\alpha}{r}$. That is, $\frac{1}{r} A(x) : \{0, 1\}^d \mapsto [0, 1]^d$. Thus, by lemma 26 we can see that $1 \geq \Omega(\frac{dr}{\epsilon \alpha})$. $\square$

*Proof of Theorem 17.* The proof is almost the same as that of Theorem 12. Assume that we have a set of data points $\mathcal{X} = \{x_1, \boldsymbol{x}_2 \cdots, x_{2^d}\} = \{0, r\}^d$. $\mathcal{A}$ will also be $\epsilon$-zCDP on $\mathcal{X}$ as in the proof of Theorem 12. Thus, if

$$\mathbb{E}[\|\mathcal{A}(x) - x\|_\infty] \leq \alpha,$$

then

$$\mathbb{E}[\|\frac{1}{r}\mathcal{A}(x) - \frac{1}{r}x\|_\infty] \le \frac{1}{r}\alpha.$$

This means that $\frac{1}{r}\mathcal{A}(x) : \{0,1\}^d \mapsto [0,1]^d$ is $\epsilon$-zCDP. Thus, by Theorem 24 we must have

$$1 \ge \Omega(\sqrt{\frac{dr^2}{\epsilon\alpha^2}}).$$

$\square$

*Proof of Theorem 18.* The proof is almost the same as that of Theorem 10. By Lemma 22, we have

$$D_\alpha(\mathcal{N}(x, \frac{dr^2}{2\epsilon})\|\mathcal{N}(x', \frac{dr^2}{2\epsilon})) = \frac{\alpha\epsilon\|x - x'\|_2^2}{dr^2} \le \frac{\alpha d\epsilon\|x - x'\|_\infty^2}{dr^2} \le \alpha\epsilon.$$

Therefore, $\mathcal{A}(x) = x + z$ with $z \sim \mathcal{N}(0, \frac{dr^2}{2\epsilon})$ is $(r, D_{MR}, \|\cdot\|_\infty, \epsilon)$-robust. The bound of $\mathbb{E}[\|z\|_\infty]$ can be easily proved by substituting $\sigma$ in $O(\sigma\sqrt{\log d})$ ((Orabona & Pál, 2015)) with $\sqrt{\frac{dr^2}{2\epsilon}}$. $\square$

## C MORE EXPERIMENTAL RESULTS (IMAGENET)

### C.1 CERTIFYING $\ell_2$ ROBUSTNESS

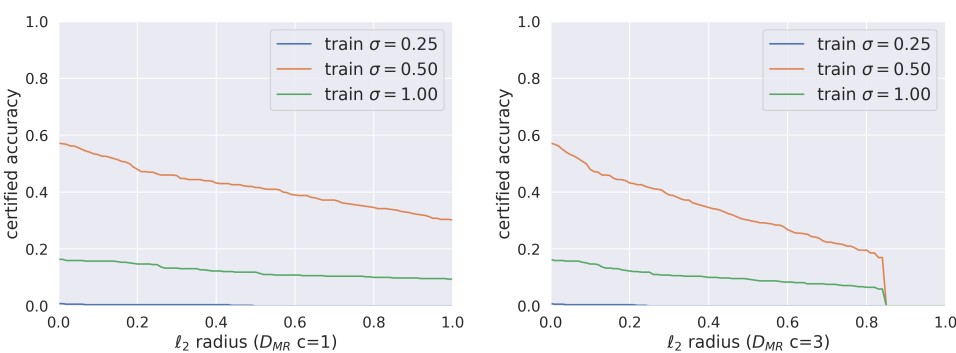

Figure 4: Certifying $D_{MR}$ robustness in $\ell_2$ norm on ImageNet: vary the the Gaussian noise in the training process and fix the $\sigma$ of the Gaussian mechanism as $\sigma = 0.5$. $c = 1$ (left) and $c = 3$ (right).

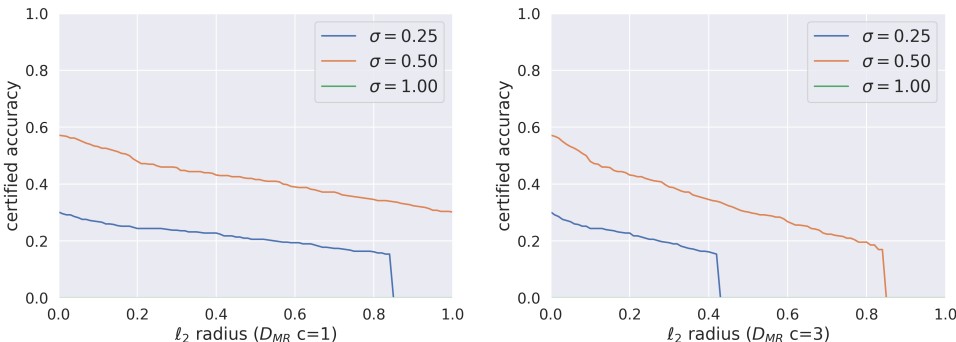

Figure 5: Certifying $D_{MR}$ robustness in $\ell_2$ norm on ImageNet: vary the $\sigma$ in the Gaussian mechanism and fix the $\sigma$ of the training noise as $\sigma = 0.5$. $c = 1$ (left) and $c = 3$ (right). *There is no green line because the accuracy is 0 when adding $\sigma = 1.0$ Gaussian noise to the images.*

### C.2 CERTIFYING $\ell_\infty$ ROBUSTNESS

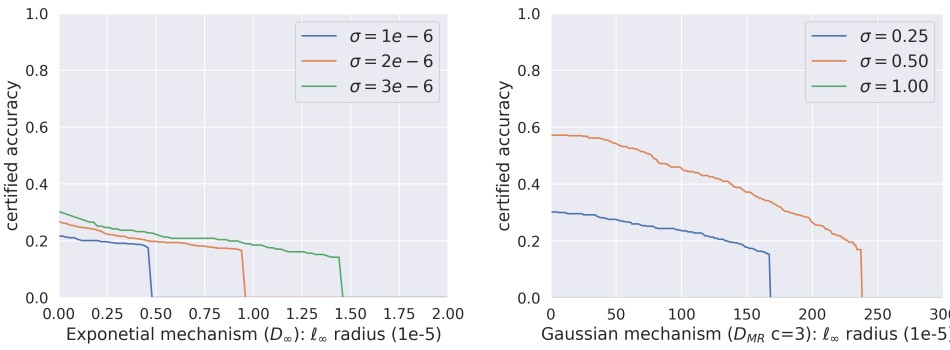

Figure 6: Certifying $D_\infty$ robustness and $D_{MR}$ robustness in $\ell_\infty$ norm by the Exponential mechanism and the Gaussian mechanism on ImageNet: vary the $\sigma$ in the exponential mechanism (left) vary the $\sigma$ in the Gaussian mechanism (right). The classifier is trained with $\sigma = 0.50$ Gaussian noise. As we can see, the certified radius is smaller than 1/255.

