# OpenReview forum: "A Unified framework for randomized smoothing based certified defenses"
_ICLR.cc/2020/Conference — Reject_

### Official Review · AnonReviewer1 · 2019-10-22
**Official Blind Review #1**

**Rating:** 3

**Review:**

This work examines the recently proposed randomized smoothing method for certifying the robustness of neural networks. The authors explain a theoretical framework for analyzing randomized smoothing as a certification method, propose two alternative definitions of robustness (D_MR and D_inf), and prove that using Gaussian noise for smoothing is near “optimal” for L2 robustness, while using exponential noise for smoothing is optimal for L_inf robustness (the authors do this by establishing a lower bound on the noise necessary for smoothing to work). This also leads the authors to the interesting conclusion that randomized smoothing may not be scalable to high dimensional data for L_inf robustness.

In its current state, I would vote to weakly reject this paper for one key reason. The notions of robustness defined by the authors (Definitions 3/7/8) is not the same as standard adversarial robustness (Definition 2), and the authors do not explain clearly how to translate their results back to adversarial robustness. Proving results about their own version of robustness is interesting, but it must be related back to the standard notion of adversarial robustness so that the broader machine learning community can understand how the authors’ contributions fit in the literature. It may in fact be quite straightforward to relate the two notions, but I think the authors should explain how to do so clearly. I am happy to reconsider if the authors can address this (and other comments below) in a satisfactory manner.

I did not check the authors’ theoretical proofs, but I find the statements of the theorems interesting, especially the results about the maximum certifiable radius for L_inf robustness. This provides significant new insight about the fact that L_inf robustness may not be easy to certify using randomized smoothing methods. However, it is not clear to me how best to translate the authors’ results to a result for the standard notion of adversarial robustness, which I believe would be interesting to present clearly.

I would encourage the authors’ to clarify (and tone down) their statement about the “optimality” of Gaussian noise for L2 robustness. Theorem 12 provides a lower bound on the L_inf norm of the noise added, and they show that Gaussian noise is close to “optimal” in terms of expected L_inf norm. I am a bit confused as to why are we providing bounds on the L_inf norm of the added noise (especially since we are verifying L2 robustness) - in what other ways is Gaussian noise (near) optimal? Does it also have the expected lowest L2 norm? Also, why do we want the noise to have low norm? I feel that “optimal” should mean being able to prove the largest possible robust radius, and if that is not what you are proving, I would encourage you to try to avoid overclaiming.

Finally, the experimental results should also not just be in terms of D_MR robustness. Otherwise, it is hard to compare with prior work like Cohen et. al.

Some additional feedback:

- “the Lp-normed robustness” can be replaced with “Lp-norm robustness” everywhere
- Page 1, say “the Gaussian mechanism” instead of “Gaussian mechanism” (toward the end of the first paragraph)
- Table 1’s formatting can be improved (maybe have a box around the whole table)
- Theorem 12 - use “In other words” instead of “In another word”

**Experience Assessment:**

I have published one or two papers in this area.

**Review Assessment: Checking Correctness Of Derivations And Theory:**

I did not assess the derivations or theory.

**Review Assessment: Checking Correctness Of Experiments:**

I assessed the sensibility of the experiments.

**Review Assessment: Thoroughness In Paper Reading:**

I made a quick assessment of this paper.

---

> ### Author Response · Authors · 2019-11-10
> **Reply to reviewer 1**
>
> Thanks for your valuable comments.
>
> Thanks for comments regarding theorem 12. You indeed make a point. In theorem 12, we provide lower bound on $L_\infty$ because we are still not sure if there is an expected $L_2$ norm lowest bound.
>
> What we prove here is that if we want to guarantee this robust radius r, how much noise should be added. From our viewpoint, *the contraposition of this claim is what you are talking about here*: adding this amount of noise, the largest possible robust radius is r.
>
> Actually , to avoid over-claim, all the main claims we make in this paper are regarding scales, and we also use phrases like "may" or "under our framework" or "certify the robustness defined in our frameworks" in the claims in the paper. *Specific to theorem 12*, at the beginning of the comments, we say "to guarantee $(r, D_{MR}, \|\cdot\|_2, \epsilon)$-robustness", which already indicates the scope of this theorem. We use "near" optimal because "up to" indicates the maximum factor is log d, and thus the factor might be small (smaller than log d).
>
>
> In the experiments, we mainly want to verify our theorems not compare the framework with the prior work since our attempt here is not to compare with the other frameworks but to study the optimality and scalability of random mechanisms.
>
>
> Thanks for your comments again.

---

### Official Review · AnonReviewer2 · 2019-10-23
**Official Blind Review #2**

**Rating:** 1

**Review:**

Summary of the paper's contributions:

This paper introduces two new notions of robustness for randomized classifiers, which are based on the notions of differential privacy (DP) of randomized mechanisms. Specifically, the D_\infty robustness and D_{MR} robustness of a random classifier are defined based on \epsilon-DP and \epsilon-zCDP, respectively.

The paper proves lower bounds on the noise level of a random classifier for which it can be certified D_\infty robust and D_{MR} robust. Further, it is shown that the lower bounds are achieved by random classifiers constructed using Gaussian noise and exponential noise for l_2 and l_\infty robustness, respectively.

Major criticisms: (1) The paper does not give sufficient motivation for studying D_\infty robustness and D_{MR} robustness. (2) The paper makes several unsubstantiated claims regarding the optimality of different noise models for adversarial robustness.

Detailed comments:

- All the claims made in the paper regarding the optimality of different noise models  are specific to D_\infty and D_{MR} robustness. However, they are written in a way to imply that the claims also hold for the standard l_2/l_\infty robustness which is studied in adversarial ML literature (especially in the abstract and intro section). The authors should make clear the relation between D_\infty and D_{MR} robustness and the standard notion of robustness of a classifier. Does one imply the other?

- Does the relaxed notion of robustness of a random classifier g (Definition 3) imply a robustness guarantee for the final output of the random classifier, i.e., \argmax_c P(g(x) = c) ?

- The experiments use the same setup as in Cohen et al, but the results are not compared with those in Cohen et al. It is not clear how to judge the significance of these results without comparison to any other method of evaluating robustness.

- "However, it is known that adding Gaussian noise often does not lead to \epsilon-DP, but rather (\epsilon; \delta)-DP (Dwork et al., 2014) which has an additional parameter \delta and thus is harder to be incorporated in our framework. To alleviate this issue, we employ Maximal Relative Renyi Divergence as the probability distance measurement to define another type of robustness, namely D_{MR} robustness." - This does not provide sufficient justification for studying D_{MR} robustness.

- A comparison is made between Theorem 10 & Corollary 11 in the paper to Theorem 1 in Cohen et al. However, it is not clear how the result in the paper is better or even equivalent to the one in Cohen et al. D_{MR} robustness seems to be an approximate notion of robustness, while the result in Cohen et al gives perfect robustness within a ball of a certain radius. The radius r in both papers scales linearly with \sigma. It is said that "a smaller c yields a larger r compared to Cohen et al." It is not clear why that is useful.

- In Theorem 12, each entry in x is restricted to be in the range [0, r/\sqrt{d}]. This means the l_2 norm of x cannot be more than r. Then, how is it meaningful to discuss the (2r, D_{MR}, l_2, \epsilon/2) robustness of an algorithm on this data, with the radius of the robust guarantee being 2r?

- In Theorem 12, the lower bound on the expected l_\infty norm of the random noise is shown to be independent of d, while the expected l_\infty norm for Gaussian noise scales with \sqrt{\log d}. I don't think it is correct to claim that Gaussian noise is "near" optimal from this analysis.

-  In Definition 23, it is not clear what Loss(Y||Y') is.

Suggestions for improvement:

- The authors should make it clear in the abstract that the optimality of the noise models is with regards to the newly defined notions of robustness.

- It would be worthwhile to discuss how D_\infty and D_{MR} robustness differ from standard notions of minimax robustness.

- One possible way to motivate the relaxed robustness introduced in Definition 3 is to link it to the robustness of the randomized classifier in Definition 1.

- Please consider using \left( \right) instead of ( ).

- For experiments, it would help to compare D_\infty and D_{MR} robustness alongside the standard l_2 robustness. In addition to training the network on Gaussian augmented dataset, it might be worthwhile to compare it to other baseline approaches as done in Cohen et al.

**Experience Assessment:**

I have read many papers in this area.

**Review Assessment: Checking Correctness Of Derivations And Theory:**

I assessed the sensibility of the derivations and theory.

**Review Assessment: Checking Correctness Of Experiments:**

I assessed the sensibility of the experiments.

**Review Assessment: Thoroughness In Paper Reading:**

I read the paper thoroughly.

---

> ### Author Response · Authors · 2019-11-10
> **Reply to reviewer 2**
>
> Thanks for your comments. We appreciate your efforts on reviewing this paper.
>
> We would like to first try to address your main criticisms before going into details.
>
> (1) Motivations for studying $D_\infty$ and $D_MR$ robustness: One big motivation of studying $D_\infty$ and $D_{MR}$ robustness is that it is easy to answer the questions in the abstract using these notions. Although D_infty and D_MR robustness are seemingly new notions, they are **actually relevant to previous "standard" notions**. We clarify the connections in the remark after theorem 8.
> Specifically, as long as $D_\infty$ robustness is certified, the expected output stability bound in (Lecuyer et al.) will be guaranteed with $\delta'=0$. And if $D_{MR}$ robustness is certified, the expected output stability bound in (Lecuyer et al.) will be guaranteed with $\epsilon'=(c+1)\sqrt{\epsilon}$ and $\delta'= \exp(-\frac{c^2}{4})$, according to Theorem 10 . Besides, the ``scale'' of the robust radius certified by our framework is similar the ``scale'' of the robust radius in (Cohen et al.), according to corollary 11.
>
> (2) Unsubstantiated claims: All the main claims we make in this paper are regarding scales, and we also use phrases like "may" or "under our framework" or "certify the robustness defined in our frameworks" in the claims in the paper. Also considering the connections between our framework and the existing ones, we are sorry that we do not think those claims are really unsubstantiated. But we still respect your points.
>
> Detailed comments:
> 1 \& 2: We clarify this in the remark after theorem 8 in the revision. You can also refer to our response to the main criticism (1). Thanks for pointing this out.
>
> 3. Our attempt here is not to compare with the other frameworks, as we mention at the beginning the "experiment" section.
>
> 4. Please refer to our response to the main criticism (1).
>
> 5. We just want to explain our framework here. As we mention in the paper, our attempt here is not to compare with the other frameworks but to study the optimality and scalability of random mechanisms.
>
> 6. We are sorry that we do not quite understand your concern regarding theorem 12. The data entries are in [0, $r/\sqrt(d)$] do not indicate their outputs are the same (similar). 2r robustness implies the outputs are similar. We are not sure if this is your question (sorry about this).
>
> 7. We clarified this claim right after it (up to a log{d} factor). We use "near" because "up to" indicates the maximum factor is log d, and thus the factor might be small (smaller than log d).
>
> 8. Loss(Y||Y') is a basic concept coming from differential privacy.
>
> Thanks for your comments again.

---

### Official Review · AnonReviewer3 · 2019-10-23
**Official Blind Review #3**

**Rating:** 3

**Review:**

Summary.
The authors propose a new definition for robustness of random functions. This definition is ideal for analyzing the certified robustness under randomized smoothing techniques. They analyze and show that the Gaussian smoothing is near optimal for \ell_2 smoothing as the mean maximum error is only off by a factor of log d where d is the dimension from the optimal mean maximum energy. This is the case even under a more strict definition of robustness defined as D_\infty. Moreover, the authors show that indeed smoothing with an exponential family  is optimal under D_\infty robustness metric with radius measured in \ell_\infty.



I find the paper very interesting and the approach is novel and generic. I do not have any major criticism.

Minor comments.
1) Equation 3 "D(A(x'),A(x))" >> "D_\infty(A(x'),A(x))"
2) Page 6 third line below Theorem 16. Reference of Theorem 11 should be Corollary 11.
3) The authors should report the certified accuracy of the undefended baseline classifier over varying radius in Figures 1 and 2 and 3.
4) Running experiments on ImageNet following Cohen et al. should make the paper stronger.
4) Can the authors comment on is the certified accuracy for \sigma=0.5 at radius = 0 is better than the unsmoothned classifier a sigma 1.0. I expect that the radius of certification is larger for larger sigma.
5) The authors should explain how does the new definition of robustness relate to the common robustness definitions as the one by Cohen et al.  More discussion is necessary for this and more justification.
6) Why is the D_MR defined as maximum over $\alpha$? It seems it is only sufficient to define it as the ratio over $\alpha$. It seems that this is only needed for Theorem 8 to hold.


------------------------------------------------------------------------

After further careful read of several relevant papers, e.g. Bun et. al 2016 and the work of Dwork "Concentrated Differential Privacy", I have several questions I would like to ask for some further clarifications.



1) Showing that a network is robust under $D_{\infty}$ robustness, implies very strong results. The type of results that are common in the literature. This is since $D_{\infty}$ robustness, implies $\epsilon$ DP networks (see Lemma 3.2 and proposition 3.3 of Bun et al.). Once $\epsilon$ DP is guaranteed identical results of Lecurer et al. can be derived immediately as this implies separation in expectation (Lecurer et al.) where one can study directly the deterministic classifier $\mathbb{E}g(\mathbf{x})$ and not the random $g$ studied in this work.

2) The authors rely on the lower bounds of Bun et al. to find the average maximum energy that preserves the $D_{\infty}$ robustness (Thm 15 and 16). Authors show that indeed exponential smoothing is optimal. This is significant but the analysis was intensively based on Bun et al.


3) The relaxation to $D_{MR}$ robustness results into improvement of the dependency on the dimension to $\sqrt{d}$ instead of $d$ for under $\ell_\infty$. This should not be surprising at all and in fact is identical to the results of Bun et al. Note that the zCDP proposed by Bun et al, is a relaxed version of DP where $\epsilon$-DP for some radius $r$ implies zCDP with radius $r^2$. See proposition 3.3. Therefore, Theorem 6 and 17 are not surprising nor are they new.


4) My major concern was with the results relating to Gaussian smoothing. I do understand that since Gaussian smoothing only implies high probability result of DP which is often referred to as ($\epsilon$,$\delta$)-DP which happens to be a equivalent to zCDP proposed by Bun et. al. Therefore, I have no issues of using $D_{MR}$ to analyzing the robustness for Gaussian smoothing since it was always analyzed in the DP community with the $\epsilon,\delta$-DP and not the stronger $\epsilon$-DP. However, the statement of the result (Theorem 12) confused me vastly. Let me clarify.


Theorem 12 seems to be too good to be true. How is it possible that one can guarantee $D_{MR}$ robustness without any dimensionality dependence. Using Gaussian smoothing the $D_{MR}$ can depend on $\sqrt{\log{d}}$. While $\sqrt{\log{d}}$ may seem small; improving this to a constant in dimension is still a very big gap from $\sqrt{\log{d}}$. This may raise several questions whether one can actually find this optimal smoothing distribution. However, with a careful read of Theorem 12, the range of the input decreases as a function of $\sqrt{d}$. That is for a given range of input (independent from d), the energy in fact is NOT constant but scales with $\sqrt{d}$. In such a case, the Gaussian smoothing is now of order $\sqrt{d \log{d}}$. Now, the factor is still $\log{d}$, but now this is very different as indeed improving the Gaussian to $\sqrt{d}$ may not be of significant interest as the energy still depends in the optimal sense on $\sqrt{d}$ which does not allow it to scale for larger problems. Moreover, Cohen et al results show that with Gaussian smoothing the energy of the noise scales $\sqrt{d}$ since the noise energy $\|n\| = \mathcal{O}(\sqrt{d} \sigma)$ where $\sigma$ is std of Gaussian. Therefore, it seems that there is nothing surprising about such a result at all. The statement of the Theorem is very misleading and confusing.

Overall, I like this new approach of analyzing the random smoothed classifier; however, the poor presentation of the work and the mis-represented Theorems that seem to over claim are a major reason for my rating. In addition, the paper should be self-contained in which one should not need to read 2-3 other works to figure out the details in this work and the meaning of the several robustness metrics and their direct relations to DP and Lecuer et al. results. The statement of constant in dimension lower bound on the energy of the noise under $D\_{MR}$ was to me the major contribution; however, I found now that the statement is misleading and that in fact it is $\sqrt{d}$ reduces the contribution of the paper particularly after learning that such lower bounds are already derived in Bun et al.

**Experience Assessment:**

I have published one or two papers in this area.

**Review Assessment: Checking Correctness Of Derivations And Theory:**

I assessed the sensibility of the derivations and theory.

**Review Assessment: Checking Correctness Of Experiments:**

I assessed the sensibility of the experiments.

**Review Assessment: Thoroughness In Paper Reading:**

I read the paper thoroughly.

---

> ### Author Response · Authors · 2019-11-10
> **Reply to Reviewer3**
>
> Thanks for your valuable comments. We appreciate your efforts on reviewing this paper.
>
> (1) Your observation on the connection between (Lecurer et al.) and our framework is correct. But actually, Eg(x) is also not a deterministic classifier. It shares the same issue with the definition of random g (Cohen et al.). Because in practice, for both definitions, we need to sample x+z (z is noise) for approximation. So all the guarantees are *probabilistic*. The connection between our framework and the lemma1 in (Lecurer et al.) also demonstrates this: Specifically, as long as $D_\infty$ robustness is certified, the expected output (Eg(x)) stability bound  in (Lecuyer et al.) will be guaranteed with $\delta'=0$. And if $D_{MR}$ robustness is certified, the expected output stability bound in (Lecuyer et al.) will be guaranteed with $\epsilon'=(c+1)\sqrt{\epsilon}$ and $\delta'= \exp(-\frac{c^2}{4})$, according to Theorem 10 .
>
> (2) The proof is based on some lemmas in Bun et al. But Bun et al. is a work on DP not on adversarial robustness.
>
> (3) This result might be not that surprising, but we are the first to prove it under a framework (that has connections with the existing frameworks). From our viewpoint, randomized smoothing is popular recently due to its strong scalability to arbitrary networks and large datasets. By this theorem, we want to show that this method might also have a limitation in terms of scalability.
>
> (4) We are sorry that we do not quite understand your concern regarding theorem 12. What we claim here is with regards to L_2 norm not L_infty norm. That's why this does not depend on dimensionality. It seems like you already understand this part.  As we mentioned above, randomized-smoothing is popular recently due to its strong scalability. Our proof here indicates it might also have a limitation in terms of scalability.  Considering the connections between our framework and the existing ones, we think this theorem at least sheds light on this question as we mention in the paper.
>
>
>
> Over-claim issue:
> The main reason we use seemingly new notions and framework is to make all the proof clearer. But there are a lot of connections between our notions and the previous work. We detail the connections in the remark after theorem 8 in the revision. To avoid over-claim, all the main claims we make in this paper are regarding scales, and we also use phrases like "may" or "under our framework" or "certify the robustness defined in our frameworks" in the claims in the paper. Therefore, we respect your points, but from our perspective, we do not think those claims really over-claim.
>
>
> Thanks for your valuable comments again.

---

### Public Comment · ~Matthew_B_Mirman1 · 2019-10-06
**IBP does not have high computational complexity**

On page 2, under related work you say:

> The second approach certifies the bounds of the outputs, given the perturbation size, using interval analysis (Mirman et al., 2018; Wang et al., 2018; Gowal et al., 2018). The main issue of the above two approaches is that they can hardly be scaled to large datasets and networks due to the associated high computational complexity.

I have a few points:

1.  Interval Analysis (IBP) does not have high computational complexity.  The forward pass is only about 8 times slower than a non analysis based forward pass.  The backward pass is roughly the same.  Training speed is usually even better than adversarial training with PGD, as often many sequential autodiff passes are necessary (if k=20, this is a guaranteed 20x slow down).

2. Its rather strange to say in general they can't scale to large datasets.  How is dataset size being measured here?  If in number of data points, usually these techniques perform better with more data.

3. Mirman et al. trains using interval analysis yes, but certifies with the Hybrid Zonotope domain.  Gowal et al. certifies with complete methods. None of these three papers mentioned train using entirely only interval analysis either.

---

> ### Author Response · Authors · 2019-10-06
> **Thanks for your comment, here we refer to *ImageNet*.**
>
> Hi, Matthew
>
> First, thanks for your comments. Here large datasets refer to datasets like ImageNet (*high dimensionality* and large data size)
>
> To my best knowledge, we have not seen any work that could certify robustness on ImageNet by IBP. A very recent work we know in this direction (IBP) certifies on tiny ImageNet. Pls correct me if our observation is wrong.
>
> To our knowledge on IBP, we should use the symbolic method to analyze very deep network and ImageNet, otherwise, the error will be very large. However, using symbolic method on very deep network and ImageNet will lead to additional computational cost (and *the error also accumulates layer by layer*).
>
> We concur with you that *this single sentence* might be a little ambiguous, so we will clarify it in the next revision (as you commented).
>
> However, *we do not think this is a (big) issue of this paper*, because what we study in this paper is *randomized smoothing* not IBP. They are completely different methods. More importantly, the main attempt of our paper is to *shed light on those questions regarding randomized smoothing* as we mention in the abstract and introduction *not compare all the methods*.  It means we even do not have to mention the details of IBP in the related work, because it is actually not very related (to our attempt).
>
> Thanks for your comment again!

---

> > ### Public Comment · ~Matthew_B_Mirman1 · 2019-10-08
> > **Both sentences are incorrect, and that *is* an issue in any scientific work.**
> >
> > > The second approach certifies the bounds of the outputs, given the perturbation size, using interval analysis (Mirman et al., 2018; Wang et al., 2018; Gowal et al., 2018).
> >
> > None of these systems certifies bounds of outputs using interval analysis.   Mirman et al uses Hybrid Zonotope and Zonotope.  Gowal et al. use complete MIP methods. Wang et al. uses linear relaxations.
> >
> > > The main issue of the above two approaches is that they can hardly be scaled to large datasets and networks due to the associated high computational complexity.
> >
> > The computational complexity of these (2 of the three methods you mentioned were training methods) methods are highly varied ranging from computationally efficient such as IBP to exponential in the worst case (MILP).   The reason most incomplete methods aren't being used on ImageNet is not because they do not scale, but because virtually all of them only provide useful bounds on networks which have undergone provability training.  While provability training is computationally efficient enough to handle large networks (and thus datasets), people have observed an experimental barrier to accuracy and provability when using it.
> >
> > For computational complexity reference:
> > DeepPoly works with 88k neurons: https://files.sri.inf.ethz.ch/website/papers/DeepPoly.pdf
> > Gowal et al. used nets with 230k neurons
> > Mirman et al. showed results on nets with up to 4.5m neurons: https://arxiv.org/pdf/1903.12519.pdf
> >
> > For contrast, an important architecture for ImageNet was AlexNet which only used 650k neurons https://papers.nips.cc/paper/4824-imagenet-classification-with-deep-convolutional-neural-networks.pdf
> >
> > Furthermore, in your response you wrote:
> >
> > > To our knowledge on IBP, we should not use the symbolic method to analyze very deep network and ImageNet, otherwise, the error will be very large.  However, using symbolic method on very deep network and ImageNet will lead to additional computational cost (and *the error also accumulates layer by layer*).
> >
> > It is possible that the results for ImageNet using IBP are suboptimal, but for CIFAR10, it looks like your l_\infty bounds are well below those obtained using IBP style training and precise 100% confidence certification methods.   For example, it looks like for even 8/255 (which for you, since you scale to [0,1] is about 0.03), your method can not certify anything, whereas symbolic methods can achieve upwards of 27% certified accuracy (you can see https://github.com/eth-sri/diffai for tables with these numbers and instructions to reproduce them).   For 2/255, your method appears to get less than 40% accuracy whereas symbolic methods can achieve upwards of 45% provability and 62% accuracy.   It is thus imperative that you fully discuss these methods, as they directly relate to randomized smoothing.

---

> > > ### Author Response · Authors · 2019-10-08
> > > **We are talking about *ImageNet*, and we will clarify those two sentences about IBP. But actually, **given the main attempt of this paper**, we do not have to compare with IBP, and we even do not have to mention more details about IBP.**
> > >
> > > Hi Mirman,
> > >
> > > First, thanks for your reply. By "this is not an issue of this paper", we mean *this is not a big issue* because what we study here is randomized smoothing not IBP. They are *completely* different methods. *The things you mention about IBP (and this direction) are not related to our theories and proof*. Moreover, the main attempt of our paper is to *shed light on those questions regarding randomized smoothing* as we mention in the abstract and introduction *not* compare all the methods. Again, **IBP is a good method but not (very) related to our main attempt, theories and proof.**
> > >
> > > Second, we just want to use "interval analysis" (from another work) to summarize those methods including MIP and linear relaxations in that sentence.  It seems not accurate to you, and we will list the methods for clarification in the related work as you commented since we know you are an expert in IBP.
> > >
> > > We do not know why you mention "our method can not certify anything (when 8/255 l-infty) here (compared with IBP)." One of our attempts is to prove that **certifying l-infty robustness by randomized smoothing might be hardly scalable**. The experiments show our proof and analysis might be correct (that is all we want to show here). **Given this attempt, we do not know why we should compare the results with IBP?** We acknowledge that IBP might perform better on cifar-10 (8/255 l-infty), but this is indeed not related to what we want to show here.
> > >
> > > As we mentioned in our last comment, **we will clarify the few sentences about IBP** (as you commented). But due to the limited space, it is difficult to clarify all the seemingly related work in the main body, and thus *we will add some background in the Appendix in our revision*. Actually, from our point of view, **differential privacy theory is even more related (than IBP)**, but we put the background in the Appendix due to the limited space.
> > >
> > > Last but not least, our attempt is to *give answers (in O()) to those questions regarding randomized smoothing under our framework* not to compare the performance and details with all the other methods. That's why we did not fully discuss *the other methods*. Actually, we even do not have to mention more details about IBP and the other methods you mention, **simply because they are not really very related to our attempt, theories, and proof**.
> > >
> > > Again, IBP is a good method, and thanks for correcting those two sentences.
> > > But IBP (and the methods you mention) are indeed not (very) related to our main attempt, theories, and proof.
> > >
> > > Thanks.

---

> > > > ### Public Comment · ~Matthew_B_Mirman1 · 2019-10-08
> > > > **Interval Analysis is not MIP**
> > > >
> > > > > Second, we just want to use "interval analysis" (from another work) to summarize those methods including MIP and linear relaxations in that sentence.  It seems not accurate to you, and we will list the methods in the related work (as you commented)... As we mentioned in our last comment, **we will clarify the few sentences about IBP** (as you commented). But due to the limited space, it is difficult to clarify all the related work in the main body, and thus *we will add some background in the Appendix in our revision*.
> > > >
> > > > It is safe to call all of DeepPoly/Wang/AI2 the "primal over-approximation certification methods."   Interval analysis only refers to the singular specific method where no relations are tracked and bounds for each operation are derived using interval arithmetic https://en.wikipedia.org/wiki/Interval_arithmetic#Simple_arithmetic
> > > >
> > > > DiffAI/IBP  are training methods and not certification methods primarily. However, DiffAI also uses primal over-approximation certification for determining certified accuracy.  IBP uses the complete method MIP for determining certified accuracy.   It is inaccurate to label MIP as interval analysis or even primal over-approximation, as it is complete.
> > > >
> > > > If you are struggling for space, why not just cite all of DeepPoly/Wang/AI2/DiffAI/IBP/Dvijotham/Raghunathan/Wong & Kolter as "research into infinite confidence deterministic certification and provable defenses" and clarify that these have the potential to overcome some of the limitations you prove apply to finite confidence probabilistic defenses.

---

> > > > > ### Author Response · Authors · 2019-10-08
> > > > > **We will refine this sentence as you commented. But since the methods you mention are not (very) related to our attempt and theories, we might try to include some details in Appendix later.**
> > > > >
> > > > > Hi Mirman,
> > > > >
> > > > > Thanks for pointing this out. We will refine this sentence as you commented. Apparently, primal over-approximation certification (as you mentioned) is another good direction. But since it is not (very) related to our attempts and theories, we might try to go through those papers and include some details in Appendix later. It is not necessary to mention the details in the main body (a clarification for other readers).
> > > > >
> > > > > Those methods (you mention) might have the potential to overcome certain limitations we prove here. However, we are not sure about it (except the well-known "probabilistic" limitation). So we prefer to study those methods meticulously (on the limitations we prove regarding randomized smoothing) later and include more details in our future work.
> > > > >
> > > > > Thanks.

---

> > > > > ### Author Response · Authors · 2019-11-13
> > > > > **Follow-up reply**
> > > > >
> > > > > Hi Mirman,
> > > > >
> > > > > Thanks for your previous comments. We appreciate you pointed those two sentences out, and we already refine one and delete one (to avoid ambiguity) in the revision based on our current understanding after reading your paper carefully.
> > > > >
> > > > > Thanks again for your valuable comments.

---

### Decision · Program_Chairs · 2019-12-19

**Decision:**

Reject

**Comment:**

After the rebuttal, the reviewers agree that this paper would benefit from further revisions to clarify issues regarding the motivation of the DP-based security definition,  any relationship it may have to standard definitions of privacy, and the role of dimensionality in the theoretical guarantees.